# Surface-Modified Inhaled Microparticle-Encapsulated Celastrol for Enhanced Efficacy in Malignant Pleural Mesothelioma

**DOI:** 10.3390/ijms24065204

**Published:** 2023-03-08

**Authors:** Xuechun Wang, Gautam Chauhan, Alison R. L. Tacderas, Aaron Muth, Vivek Gupta

**Affiliations:** 1Department of Pharmaceutical Sciences, College of Pharmacy and Health Sciences, St. John’s University, 8000 Utopia Parkway, Queens, NY 11439, USA; 2Department of Biological Sciences, College of Liberal Arts and Sciences, St. John’s University, 8000 Utopia Parkway, Queens, NY 11439, USA

**Keywords:** celastrol, surface wrinkling, microparticle, pulmonary delivery, malignant pleural mesothelioma, PLGA

## Abstract

Malignant pleural mesothelioma (MPM) is a rare and aggressive cancer affecting the pleural lining of the lungs. Celastrol (Cela), a pentacyclic triterpenoid, has demonstrated promising therapeutic potential as an antioxidant, anti-inflammatory, neuroprotective agent, and anti-cancer agent. In this study, we developed inhaled surface-modified Cela-loaded poly(lactic-co-glycolic) acid (PLGA) microparticles (Cela MPs) for the treatment of MPM using a double emulsion solvent evaporation method. The optimized Cela MPs exhibited high entrapment efficiency (72.8 ± 6.1%) and possessed a wrinkled surface with a mean geometric diameter of ~2 µm and an aerodynamic diameter of 4.5 ± 0.1 µm, suggesting them to be suitable for pulmonary delivery. A subsequent release study showed an initial burst release up to 59.9 ± 2.9%, followed by sustained release. The therapeutic efficacy of Cela MPs was evaluated against four mesothelioma cell lines, where Cela MP exhibited significant reduction in IC_50_ values, and blank MPs produced no toxicity to normal cells. Additionally, a 3D-spheroid study was performed where a single dose of Cela MP at 1.0 µM significantly inhibited spheroid growth. Cela MP was also able to retain the antioxidant activity of Cela only while mechanistic studies revealed triggered autophagy and an induction of apoptosis. Therefore, these studies highlight the anti-mesothelioma activity of Cela and demonstrate that Cela MPs are a promising inhalable medicine for MPM treatment.

## 1. Introduction

Malignant mesothelioma (MM) is a rare, incurable, and malignant type of cancer arising from the mesothelial or sub-mesothelial cells of the pleura (lung lining), peritoneum (abdominal lining), pericardium (heart sac), or testes [1]. MM is classified as a rare disease with ~30,000 new cases diagnosed worldwide annually [2], with a subtype, malignant pleural mesothelioma (MPM), accounting for 80–90% of all MM cases. Occupational exposure to asbestos fibers has been identified as the major cause of MPM in patients, leading to various symptoms, such as difficulty breathing, chest pain, fluid buildup, and others [3,4]. Due to a latency period in tumor development of 20–50 years after exposure to asbestos, the incidences of MPM have increased slightly in the last decade, especially in industrialized countries, such as the UK, Australia, and Belgium [1,3]. Additionally, roughly 3000 new cases are diagnosed each year in the United States [5]. Current therapies include surgical resection or combinations of chemotherapy, radiation, and immunotherapy; however, these approaches have not shown great success with patients experiencing multiple adverse side effects and only give a ~10% 5-year survival rate [6]. Since MPM has been studied less extensively than other primary neoplasms of the chest, there is an increased urgency to explore and develop more efficacious treatment options while minimizing side effects.

Celastrol (Cela) is an extensively studied phytochemical derived from the root extract of *Tripterygium wilfordii Hook F (Thunder God Vine)*, which has demonstrated interesting and promising bioactivity [7]. Numerous studies have demonstrated Cela’s intriguing pharmacological properties, including anti-inflammation [8,9], neuroprotection [10,11], and anti-cancer [12,13]. Additionally, Cela has been shown to induce apoptosis in cancer cells by activating apoptotic proteins such as caspase-3/8/9 [14,15]. Cela has also been reported to induce autophagy and cause tumor growth inhibition [16,17]. Despite encouraging in vitro and preclinical efficacy, the clinical applications of Cela remain challenging due to its extremely low aqueous solubility, low bioavailability, narrow therapeutic window, and potential toxicity [7,12]. Therefore, the use of an effective carrier may help to improve these unfavorable physiochemical and pharmacokinetic properties and possibly enhance its treatment potential while reducing off-target cytotoxicity. 

Pulmonary drug delivery presents various advantages over oral and parenteral routes for localized treatment of chronic lung disorders. This route of administration provides targeted lung delivery, local accumulation, rapid onset, reduced systemic exposure, extensive surface area for drug absorption, and reduced side effects, and requires lower doses for similar lung deposition of therapies [18,19]. The aerodynamic diameter of inhaled particles should range from 2–5 µm for efficient deep lung targeting [20]; however, the entry of foreign materials into the respiratory tract may activate defense mechanisms which may remove them (e.g., mucociliary clearance and alveolar macrophage phagocytosis) [21]. Recently, polymer-based porous microparticles have been identified as effective carriers for pulmonary delivery due to their large geometric diameters and low densities, which not only provide protection from clearance by macrophages, but are also capable of being deposited in the deep lung [19,22,23,24]. A study by Hu et al. developed inhalable curcumin-loaded poly(lactic-co-glycolic) acid (PLGA) porous microparticles with a mean geometric diameter >10 µm and an aerodynamic diameter of only 3.12 µm, which effectively avoided uptake by alveolar macrophages [23]. PLGA has been widely used to develop porous microparticles for delivering small molecules, proteins, and plasmid DNA to the lungs [25,26,27,28,29]. 

This is the first report demonstrating Cela’s efficacy for MPM treatment. In addition, the present study explores surface-modified microparticles as effective inhaled carriers possessing deep lung deposition properties. Cela-loaded PLGA microparticles (Cela MPs) were prepared using pore-forming agent sodium chloride (NaCl) to create corrugated surface morphology that provides favorable aerodynamic properties while efficiently avoiding macrophage uptake. Optimized Cela MPs were analyzed for their aerodynamic performance as well as their 2D anti-proliferative properties and 3D anti-tumor efficacy against mesothelioma cell lines. Finally, multiple anti-cancer mechanisms of action of Cela MPs were elucidated. 

## 2. Results

### 2.1. Formulation of Celastrol Microparticles (Cela MPs)

Cela-loaded microparticles were successfully prepared using a double w/o/w emulsion solvent evaporation method (Table 1 and Appendix A). The F1 formulation resulted in a good %EE of 63.7% with a negative of −30.3 mV. To increase zeta potential to potentially improve cell internalization, we introduced 1% PEI in the inner aqueous phase and obtained an F2 formulation with a 39.5 mV zeta potential and ≈60% drug entrapment. Formulations F3 and F4 consisted of the same parameters as F1 and F2, respectively, with the exception that the outer aqueous phase was increased to 2% PVA. The resulting %EEs (F3: 67.1% and F4: 52.0%) and zeta potential (F3: −32.9 mV and F4: 35.7 mV) were found to be similar to that of the F1 and F2 formulations. Therefore, 1% PVA was selected for the latter formulations. Comparatively, the addition of PEI in the composition (F2 and F4) reduced the %EE; therefore, PEI was not included in further formulations. In this project, we aimed to create large porous particles to achieve the required aerodynamic size for delivery and to reduce the risk of phagocytosis [30]. Two pore-forming agents, or porogens (NaHCO_3_ and NaCl), were examined. Incorporating NaHCO_3_ (F5) in the inner aqueous phase resulted in a reduced %EE of 34.8%, particle size of 2341 nm, and zeta potential of −21.6 mV, along with a very high PDI of 0.791. However, the addition of NaCl (F6) as a porogen resulted in a relatively higher %EE of 72.8 ± 6.1%, lower PDI of 0.3 ± 0.2, and a stable zeta potential of −36.6 ± 3.8 mV, along with a particle size of 2076 ± 390 nm geometric size and a PDI of 0.3 ± 0.2. F6 formulation was selected for further studies.

### 2.2. Morphological Analysis

The morphology of Cela MPs was examined by SEM. A representative image is presented in Figure 1A, which reveals a wrinkled pattern that covers the surface of the MPs and is ~2 µm in diameter. The fold-like patterns on the surface are due to the addition of NaCl during the preparation process that is responsible for the disruption of the PLGA matrix as the MPs are formed.

### 2.3. In Vitro Release Study

The release profile of Cela MPs in 1% SLS in SLF at 37 °C is shown in Figure 1B. As can be seen, after 30 min, 59.9 ± 2.9% of Cela was released, suggesting an initial burst release from the MP formulation, followed by sustained release up to 81.0 ± 1.0% by 72 h. This demonstrates the ability of MPs to release a majority of the entrapped drug soon after reaching the lungs, followed by maintenance doses being released over a period of time. Several reports have suggested the elimination half-life of Cela being 7.5–10 h in rats, which outweighs the need to develop extended-release dosage forms [31].

### 2.4. Differential Scanning Calorimetry (DSC) Studies

The DSC thermogram of Cela (Figure 1C) shows a sharp endothermic peak at 155 °C and an exothermic peak at 211 °C. Both physical mixture samples displayed endothermic and exothermic peaks for Cela at 156 °C and 210 °C, respectively, with an additional endothermic peak originating from the amorphous PLGA polymer at ~50 °C [32]. However, these characteristic peaks were absent in the thermograms of the blank MP and Cela MP, which indicated the successful encapsulation of Cela in the microparticles.

### 2.5. X-ray Diffraction Studies (XRD)

Diffractometry studies revealed multiple peaks representing the crystalline nature of Cela, shown in Figure 1D. The diffractograms of the physical mixtures comprising of Cela + PLGA + NaCl and Cela + Blank MP both clearly demonstrate the presence of Cela crystalline peaks. Upon analyzing the Blank MP and Cela MP, Cela peaks were absent over the entire range of 10–80° 2θ values. The results obtained from the XRD studies also indicate successful encapsulation of Cela in the microparticles.

### 2.6. In Vitro Stability

The stability of optimized Cela MP was determined in terms of particle size, zeta potential, and percent drug entrapment as a function of storage time and conditions. As seen in Figure 2, there was no significant change in all three parameters after the Cela MP suspension was stored for 28 days at 4 °C. The particle size at day 28 was 2162.7 ± 376.1 nm, compared to 2358.3 ± 359.6 nm at day 0 (Figure 2A). The zeta potential at day 0 was −24.4 ± 2.5 mV, and at day 28 it was −25.2 ± 3.3 mV (Figure 2B). The %EE at day 0 was 84.7 ± 2.1%, and on day 28 it was 90.1 ± 5.7 mV (Figure 2C). The in vitro stability studies performed for Cela MP suspension found it to be stable at 4 °C for a period of 28 days.

### 2.7. Aerosolization of Cela MPs

In vitro lung deposition studies were performed using a Next Generation Impactor (NGI). Figure 3A shows the deposition of microparticles at various stages of the NGI. As can be seen, most of the emitted particles were deposited in stage 3 and below, representing the bronchi-alveolar region. Figure 3B demonstrates the percent cumulative deposition of Cela MPs as a function of effective cut-off diameter. As can be seen, over 60% of MPs possessed a cut-off diameter of 3.3 µm, which is equivalent to stage 4 or below on the NGI (Figure 3B). The mass median aerodynamic diameter (MMAD) of Cela MP (4.5 ± 0.1 µm) was within the range of a required aerodynamic diameter for sufficient delivery to the respirable region of the lungs, i.e., 2–5 µm (Figure 3C). The GSD was calculated to be 2.1 ± 0.1 µm, indicating the aerodynamic size range of MPs from the MMAD value. The FPF, or respirable particles with aerodynamic diameter from stage 3 (5.39 µm) of the NGI and below, was 85.3 ± 1.5%, and the amount of residual formulation left in the cup after nebulization period (percent device) was 51.8 ± 6.4%, as shown in Figure 3C. This aerosolization data suggests that the Cela MP can be efficiently delivered to the lungs.

Additionally, %EE, PS, PDI, and zeta potential were evaluated for Cela MPs after nebulization. As presented in Appendix A, all four parameters showed no significant difference from the data presented for the optimized formulation in Table 1. Therefore, Cela MPs maintained their stability after nebulization.

### 2.8. Cytotoxicity Studies

The cytotoxicity of Cela MP was compared to Cela only against four mesothelioma cell lines using an MTT assay (Figure 4). As can be observed, Cela MP significantly improved the cytotoxic activity of Cela against all four cell lines (Figure 4A–D), with significantly more robust cytotoxic activity in H28 and H2452 cells. The IC_50_ values of Cela and Cela MPs were calculated using GraphPad Prism^®^, and it was observed that microparticle encapsulation reduced the IC_50_ of Cela ~1.3-fold against MTSO-211H cells (Cela: 3.6 ± 0.4 µM vs. Cela MP: 2.7 ± 0.3 µM; *p* < 0.05) (Figure 4A; Table 2). Similarly, IC_50_ values were reduced by ~3.4, ~5, and ~1.5-folds against H28, H2452, and ROB cells, respectively (Figure 4B–D and Table 2).

To determine whether blank MPs imparted any notable toxicities, blank MPs were evaluated against MSTO-211H cells and normal human lung fibroblast (NHLF) cells. As shown in Figure 5, blank MPs did not induce any appreciable toxicity, and the cell viability remained ~100% for all concentrations evaluated, in both MSTO-211H mesothelioma cell lines (Figure 5A) and NHLF normal primary lung fibroblasts (Figure 5B). This highlights the improvement in Cela efficacy being due to microparticle encapsulation and not on other formulation components.

### 2.9. Mechanistic Studies

#### 2.9.1. DPPH Assay

Cela is a known antioxidant that may boost its anti-inflammatory and anti-cancer properties. To test the ability of Cela and Cela MPs in scavenging free radicals, a DPPH assay was performed. As shown in Figure 6A,B, the DPPH radical scavenging activity of Cela and Cela MPs dose-dependently increased using 0.5, 1, 5, and 10 µM concentrations. After 30 min, the antioxidant activity of Cela only and Cela MPs was very similar (Figure 6A). After 24 h, the antioxidant activity of Cela MPs surpassed that of the Cela solution (Figure 6B). For example, 5 µM Cela exhibited scavenging activity of 15.7 ± 1.9%, while Cela MP resulted in scavenging activity of 30.2 ± 4.1%. The antioxidant effects were statistically significant at higher concentrations (5 and 10 µM, *p* < 0.0001). These results suggest that the antioxidant activity of Cela was either retained or enhanced upon encapsulation into the PLGA microparticles.

#### 2.9.2. Caspase-3 Assay

Caspase-3 activity was then monitored to examine the effects of Cela and Cela MP treatment on apoptotic pathways. Figure 6C displays the percent fluorescence intensity of caspase-3 after MSTO-211H cells were treated for 6 h. As can be seen, compared to the control (100%), lower concentration (0.5 µM) treatments resulted in no significant difference in caspase-3 levels (Cela: 102.1 ± 6.2%; Cela MP: 218.1 ± 99.5%; ns). Conversely, treatment of 1.0 µM Cela MP (503.1 ± 93.1%) resulted in a ~5-fold increase in caspase-3 levels compared to the control (*p* < 0.01) and ~3.5-fold increase compared to 1.0 µM Cela (143 ± 19.7%) (*p* < 0.01) (Figure 6C). Therefore, a significant apoptotic induction was observed with Cela MP-treated cells as compared to the drug only-treated cells, suggesting induction of apoptosis as a mechanism for its anti-cancer properties.

#### 2.9.3. Effect of Cela and Cela MP on Cellular Autophagy

Cela has also been reported to trigger autophagic pathways in which promote cell death in cancer [33,34,35]. In this study, an in vitro CYTO-ID^®^ Autophagy Detection Kit was used to determine any effect on LC3 levels, where increased levels indicated induced autophagy. The effects of Cela and Cela MP were investigated in the MSTO-211H cell line. As shown in Figure 6D, the LC3 fluorescent signal upon treatment with 5 µM Cela MP increased significantly, ~1.3-fold relative to the control cells (*p* < 0.01) and ~1.5-fold relative to Cela (*p* < 0.001). No significant difference was found between the control and Cela only treatments. This signifies that Cela MP induced autophagosome formation in mesothelioma cells at a concentration of just 1.0 µM.

### 2.10. Determination of Cela MP Efficacy in 3D Spheroid Model

#### 2.10.1. Tumor Volume Reduction Studies

An in vitro tumor simulation model was utilized to mimic in vivo tumor conditions and better predict the clinical effectiveness of Cela MP. The single-dose study involved drug treatment once throughout the 15-day experimental period. Representative tumor spheroid images from single dose treatment are shown in Figure 7A. As can be seen, treatment with both 0.5 and 1.0 µM Cela MPs visibly inhibited spheroid growth as compared to the control and Cela only at the same concentrations (Figure 7A). A detailed analysis of the spheroid volume (normalized to day 0 volume), as represented in Figure 7B, indicated that after 15 days of treatment, a lower concentration of Cela MP (0.5 µM) significantly suppressed spheroid growth by ~1.6-fold (*p* < 0.01) as compared to the control. Additionally, a higher concentration of Cela MP (1.0 µM) suppressed growth by ~13-fold as compared to the control (*p* < 0.0001) and ~10-fold as compared to 1.0 µM Cela only (*p* < 0.0001).

A multiple-dose study was then performed to mimic the physiological conditions a treatment may undergo (e.g., drug metabolism and clearance) [36,37]. Similarly, Cela MP in the multiple-dose studies showed superior anti-cancer activity as compared to the drug only (representative spheroid images from each group shown in Figure 8A). In the multiple-dose studies, all four treatment groups displayed significant suppression of spheroid growth as compared to the control (Figure 8B) (0.5 µM Cela: ~1.5-fold; 0.5 µM Cela MP: ~7.6-fold; 1.0 µM Cela: ~3.5-fold; 1.0 µM Cela MP: ~11.7-fold). However, at a concentration of just 0.5 µM, Cela MP-treated spheroids were significantly reduced in size as compared to those treated with Cela only (~5-fold reduction) (*p* < 0.0001). Therefore, the 3D spheroid volume data concludes that enhanced anti-tumorigenic efficacy is achieved for our formulation against MPM.

#### 2.10.2. Cell Viability Assay

A CellTiter Glo assay was also performed to determine the cytotoxic potential of the previously described spheroid treatments. Figure 7C represents the percent cell viability of single-dosed spheroids relative to the control. As can be seen, only 1.0 µM Cela MP treatment showed a significant reduction in cell viability as compared to the control (~9-fold reduction) and to 1.0 µM Cela only (~8-fold reduction) (*p* < 0.0001). The multiple-dose treatment regimen proved that all four treatment groups significantly reduced cell viability as compared to the control (*p* < 0.0001) and no statistical difference was found between the treatment groups (Figure 8C). The data obtained from the CellTiter Glo assay demonstrated that a single dose of 1.0 µM Cela MP can effectively penetrate the 3D spheroid and reduce cell viability.

#### 2.10.3. Live/Dead Assay

A Live/Dead assay helps visualize the viable and necrotic cells that are present in the treated spheroids. As shown in Figure 9A, green fluorescence (GFP) represents live cells, whereas red fluorescence (RFP) represents dead cells. As can be seen, a single dose of 1.0 µM Cela MP increased RFP intensity as compared to the control and other treatments. Multiple-dose spheroid images showed all four treatments reduced GFP intensity as compared to the control. Figure 9B,C represent quantitative comparisons of RFP intensity relative to the control spheroids. Quantification of the single dose study indicated that 1.0 µM Cela MP elevated RPF intensity by ~2.5-fold as compared to the control and 1.0 µM drug only. The multiple-dose data showed no significant difference in RFP intensity as compared to the control. This may be due to the decreased size of the treated spheroids as compared to the control (as mentioned in Section 2.9), thereby also resulting in reduced RFP intensity.

## 3. Discussion

Celastrol (Cela), a natural product from *Tripterygium wilfordii* Hook F (Thunder God Vine), is known to be efficacious against a variety of cancers, including liver cancer, breast cancer, prostate cancer, multiple myeloma, glioma, etc. [38]. This project presents the first report of Cela’s efficacy for MPM treatment. Studies suggest that the anti-cancer properties of Cela can be attributed to: (i) induced apoptosis and autophagy, (ii) cell cycle arrest, (iii) anti-angiogenic activity, (iv) anti-inflammatory activity, and (v) antioxidant properties [38,39,40]. However, the clinical use of Cela has been limited by multiple factors: poor water solubility, low bioavailability, narrow therapeutic window, and undesired side effects [41,42]. To address these issues, we aimed to develop a microparticle carrier system for Cela with high aerosolization efficiency to reduce the effective dose and enhance the therapeutic efficacy of the drug for MPM treatment.

The optimized Cela MP (F6 formulation) resulted in high drug entrapment (72.8 ± 6.1%), which is equivalent to 1.5 ± 0.12 mg (or 3.2 ± 0.3 mM), which represents a significant improvement in Celastrol solubilization and encapsulation of about ≈0.8 mM, as shown by Shukla et al. using cyclodextrin complexation [12]. The zeta potential of the MPs was found to be below −30 mV, which indicated stable nanofluids outside of the +30 and –30 mV range [43]. The geometric size of the MPs was analyzed to be 2076 ± 390 nm, which is within the range of 1–5 µm which is desired for the best delivery efficiency [44]. However, this particle size range was also previously shown to be ideal for phagocytosis, as per a study by Champion et al. [45]. Therefore, to circumvent this limitation, we developed microparticles with irregular surfaces using NaCl as the porogen. According to Yang et al., highly porous PLGA microparticles were able to avoid phagocytosis by macrophages, while non-porous small particles were quickly taken up by macrophages [46]. As observed from the SEM image, the MP is ~2 µm in diameter and possesses a wrinkled surface. Sodium chloride was used to create an osmotic gradient capable of driving water into the internal droplets, thus causing them to swell in size. During the evaporation process, the swollen inner droplets are immobilized within the PLGA matrix and are eventually evaporated, ultimately creating pores [47]. However, our formulation resulted in wrinkled surfaces instead of a porous structure. Based on mechanics described by Li et al., we theorize that the use of a very small amount of NaCl (1% of inner aqueous phase) caused the inner droplets to not swell enough as to adequately form pores in the PLGA matrix [47,48]. Additionally, during the evaporation process, the particles shrank as the inner droplets were removed. As shrinkage reaches a critical point, the particle surface bifurcates into fold-like structures to release the circumferential compression within the shell or elastic strain energy [48]. Therefore, the particle size of the MPs was not observed to be above 5 µm. Furthermore, studies have reported wrinkled particles prevent friction, interlocking forces, and water bridge formations due to smaller distances between the particles [49], as well as promote cell attachment without any chemical processing as compared to spherical particles [50].

Release of the drug from the microparticles is important to predict the in vivo performance of the formulation. The developed formulation displayed a biphasic release pattern characterized by an initial burst release due to the drug being present near the particle surface [51], followed by a sustained release that may be attributed to gradual degradation of the PLGA matrix and diffusion of Cela through the matrix [52,53]. This release profile proved to be highly effective at increasing the cytotoxic effect of Cela MP against MPM cells. In addition, Cela has shown to exhibit a long elimination half-life (more than 7 h) in rats; therefore, a sustained-release formulation may not be necessary to obtained the desired therapeutic efficacy of Cela [31].

DSC studies showed that Cela had a characteristic endothermic peak at 155 °C, corresponding to its melting point, and an exothermic peak at 211 °C, likely due to degradation of Cela [54]. Both DSC and XRD studies observed the absence of characteristic Cela peaks in the Cela MP sample, suggesting successful inclusion of Cela in the microparticle formulation. To design a successful delivery system, it was necessary to ensure a stable microparticle formulation. Since it is expected that the MP formulation will be stored in the refrigerator prior to use, the stability study was carried out at 4 °C over 28 days. Subsequently, it was demonstrated that Cela MPs maintained insignificant changes in particle size, zeta potential, and %EE, proving the polymers maintained the structural integrity of the MPs during long-term storage.

Conventional routes of administration, such as oral and intravenous, are not ideal for Cela delivery. For instance, a pharmacokinetic study done on rats demonstrated the mean oral absolute bioavailability after oral administration (3 mg/kg) was very low at 3.14% [55]. Even though bioavailability is not a concern in intravenous administration, Cela is associated with many side effects, including infertility, cardiotoxicity, hepatotoxicity, hematopoietic system toxicity, and nephrotoxicity [38]. Thus, the pulmonary route of administration is inarguably the best choice to efficiently deliver therapeutics locally to the lungs to improve bioavailability and avoid adverse effects. In a recent report, inhaled nintedanib (given at 1:120 of the oral dose) was found to deliver an oral-equivalent lung C_max_ with lower systemic AUC, and was well-tolerated and effective at reducing bleomycin-induced pulmonary fibrosis [56]. However, simple pulmonary delivery of free drug suspension is insufficient for deep lung deposition due to the large crystalline drug particles and variability in particle sizes. Therefore, effective pulmonary drug delivery can be achieved by optimizing the physical properties of the particle formulation, including size, charge, density, shape, solubility, and lipophilicity [57]. NGI is a popular and competent tool to evaluate the aerodynamic properties of particles based on their deposition profile throughout the stages, and was used to evaluate the wrinkled-surface Cela MPs [52,58,59]. A MMAD of 4.5 ± 0.1 µm suggested that Cela MPs are within the ideal range of 1–5 µm for the greatest pulmonary delivery efficiency [60]. Multiple studies have demonstrated that wrinkle morphology enhances the aerodynamics of aerosol in inhalation purposes [61,62,63]. Therefore, Cela MP is a promising strategy to ensure efficient delivery to the respirable regions of the lung. Additional characterization studies performed on Cela MPs after nebulization demonstrated that the MPs maintained their original %EE, PS, PDI, and zeta potential. Thus, Cela MPs are suitable for the pulmonary route of administration.

The in vitro cytotoxicity studies demonstrated that microparticles can be used as prospective carriers for MPM treatment by improving therapeutic efficacy at reduced doses. This may be explained by better cell attachment that may cause enhanced cellular uptake and subsequent cytotoxicity. Particles with wrinkled surfaces are widely found in nature, such as plant pollens and microorganisms (i.e., neutrophils). These wrinkles with their significantly enlarged surface areas provide enhanced survival tools, including pollen adhesion and hydration and cell signaling [64,65]. Inspired by these irregular morphologies, a study by Li et al. prepared wrinkled particles that cells readily attached, climbed, and conformed onto, which was not observed on the smooth particles [50]. Cell attachment to the surface of the particles was observed with actin networks appearing at the particle edges [50]. These findings provide explanation for our observations of Cela MPs performing significantly better than free Cela. Furthermore, the toxicity study of the blank MPs was evaluated for MSTO-211H cells and NHLF. The blank MPs displayed non-significant cell toxicity, suggesting the cytotoxic effects of Cela MPs are due to Cela encapsulation into the carrier system, not from other formulation components.

The antioxidant activity of Cela has been reported in various diseases [66,67,68]. Previous studies have demonstrated that Cela enhanced the activity of enzymatic antioxidants (superoxide dismutase (SOD), catalase (CAT), glutathione peroxidases (GPx), and glutathione reductase (GR)) in bleomycin-induced pulmonary fibrotic rats [69]. Another study by Wang et al. found that Cela markedly increased the activities of SOD and GPx while also decreasing levels of reactive oxygen species and MDA in obese rat models [70]. In order to evaluate the antioxidant activity of Cela MP, a DPPH radical assay was performed to determine the capacity of the drug/formulation to react with free radicals [71]. After 30 min, Cela MP showed a similar radical scavenging effect to Cela only. However, after 24 h, the antioxidant activity of Cela MP was significantly greater than Cela. This can be explained by the ability of the microparticles to encapsulate and protect Cela from the DPPH solution, thereby keeping the drug stable and sustaining its release from 30 min to 24 h while continually scavenging DPPH free radicals.

Caspase-3 is a cysteine protease that plays a vital role in the terminal course of programmed cell death (apoptosis). As a proteolytic enzyme, caspase-3 is responsible for the cleavage of DEVD peptide, poly(ADP-ribose) polymerase (PARP), DNA-dependent protein kinase, etc. [72,73]. Failure to activate apoptotic pathways in response to drug treatment may lead to drug resistance in tumor cells. In this study, 1.0 µM Cela MP was capable of inducing high levels of Caspase-3, indicating the induction of apoptosis in cancer cells. Conversely, autophagy is a cellular degradation process that occurs under stressful conditions in adaptation to starvation, development, cell death, and tumor suppression [74,75]. Autophagy is mediated by the formation of autophagosomes that collect degraded components and then fuse with lysosomes to be recycled [76]. The modulation of autophagy plays a role in both tumor suppression and promotion of many cancers, including MPM [4,76,77,78]. This study found that 1.0 µM Cela MP triggered autophagy by observing elevated fluorescent levels of LC3; however, plain Cela did not show any significant difference as compared to the control. A previous study by Liu et al. confirmed these findings, where Cela increased the formation of autophagosomes and accumulation of the LC3B-2 protein in glioma cells [79]. Li et al. also found that Cela increased levels of LC3B-2 in human osteosarcoma cells, and further observed that autophagy mediated by Cela promoted a pro-death function in cancer cells [35].

While the in vitro assays performed provided great insight into the efficacy of Cela MPs, they were likely inadequate in fully predicting the preclinical behavior of MPs. Tumors in the human body exist as a solid mass of cells that proliferate uncontrollably and grow exponentially into a three-dimensional structure. The in vitro monolayer studies lack the behavioral properties of a solid tumor, including cell-cell interactions, cellular heterogeneity, spatial architecture, and establishment of unique gene expression patterns [52,80]. Therefore, the efficacy of Cela MP was evaluated using a 3D spheroid model to mimic in vivo conditions. This study acts to bridge the gap between in vitro and in vivo studies. Spheroid volume analysis results found that Cela MP exhibited superior anti-tumor activity as compared to Cela only. Interestingly, a single-dose of Cela MP at 1.0 µM was sufficient to entirely inhibit spheroid growth after the 15-day period. To corroborate these results, we performed a cell viability assay and a live/dead cell fluorescence assay at the end of treatment period to further understand how Cela MPs suppressed spheroid growth. The results from both assays confirmed that a single dose of 1.0 µM Cela MP was effective in suppressing spheroid growth as compared to free Cela. In conclusion, results from the current study illustrate the great potential of Cela MP for the treatment of MPM.

## 4. Materials and Methods

### 4.1. Materials & Cell Lines

Celastrol (Cela) was purchased from Adooq Bioscience (Irvine, CA, USA); Resomer^®^ RG 502H (Poly(D,L-lactide-co-glycolide 50:50) (PLGA; MW 7000–17,000 Da, acid terminated) was purchased from Evonik Corporation (Piscataway, NJ, USA). Branched polyethyleneimine (PEI; average MW ~25,000 Da), Poly(vinyl alcohol) (PVA) (Mw 13,000–23,000 Da, 87–89% hydrolyzed), and 2,2-Diphenyl-1-picrylhydrazyl (DPPH) were purchased from Sigma-Aldrich (St. Louis, MO, USA). Sodium chloride (NaCl), sodium bicarbonate (NaHCO₃), 3-(4,5-dimethylthiazol-2-yl)-2,5-diphenyltetrazolium bromide (MTT), dichloromethane (DCM), dimethyl sulfoxide (DMSO), acetonitrile (ACN), other UPLC grade chemicals, and LC-MS grade water were purchased from Fisher Scientific (Hampton, NH, USA). Molecular biology kits and supplies were acquired from other commercial vendors as listed at suitable places throughout the manuscript.

Three immortalized malignant pleural mesothelioma cell lines, MSTO-211H, H28, and H2452 were procured from American Type Culture Collection (ATCC; Manassas, VA, USA). One primary patient-derived mesothelioma cell line, ROB, was obtained from Dr. Raffit Hassan at the National Cancer Institute (Bethesda, MD, USA) under an executed material transfer agreement and approved IRB protocol # 0818-034 from St. John’s University. This primary cell line was isolated from neoplastic effusions of patients undergoing therapeutic paracenteses, originally provided by The Stehlin Foundation (Houston, TX, USA) to the NIH, and were established as ROB in a study by Li et al. [81]. Normal human lung fibroblasts (NHLF) were obtained from Lonza (Basel, Switzerland). All mesothelioma cell lines were cultured in RPMI-1640 medium supplemented with 10% fetal bovine serum (FBS) (R&D Systems, Minneapolis, MN, USA), 1% sodium pyruvate, and 1% penicillin/streptomycin (Corning, NY, USA). NHLF cells were cultured in FBM^TM^ basal medium supplemented with fibroblast growth medium (FGM^TM^)-2 SingleQuots^TM^ supplements from Lonza. All cell lines were incubated at 37 °C/5% CO_2_ and 90–100% relative humidity. 

### 4.2. UPLC Method for Celastrol (Cela) Content

A reverse-phase liquid chromatography technique was established for measuring Cela content using a Waters Acquity series UPLC (Waters, Milford, MA, USA). The column used was an XBridge^®^ BEH Shield RP18 2.5µm (3.0 × 100 mm) (Waters, Milford, MA, USA). The mobile phase was 0.1% orthophosphoric acid (OPA) in HPLC grade water: acetonitrile (ACN) at a 10:90 ratio with a flow rate of 0.8 mL/min. The wavelength used for detection was 425 nm and data were collected and processed using EMPOWER 3.0 software (Waters, Milford, MA, USA). A representative UPLC chromatogram of Cela is shown in Appendix A.

### 4.3. Formulation of Celastrol Microparticles (Cela MPs)

Cela MPs were fabricated using a double emulsion solvent evaporation method previously published with some modifications [71]. Specifically, 1% NaCl (porogen) was dissolved in 0.5 mL of milli-Q water as the inner aqueous phase. The organic phase consisted of 2 mg of Cela and 60 mg of PLGA dissolved in 20 µL of DMSO and 3 mL of DCM. The primary emulsion was formed by probe sonicating (QSONICA-Q500, QSonica, Newtown, CT, USA) the inner aqueous phase with the organic phase for 1 min at 20% amplitude and 10 s on-off cycles. The primary emulsion was added to 10 mL of a 1% *w*/*v* PVA solution and homogenized (Homogenizer 850, Fisher Scientific, Hampton, NH, USA) for 1 min at 8000 rpm to obtain the double emulsion (*w*/*o*/*w*), followed by overnight stirring to remove DCM. The next day, smaller particles remaining in the supernatant were removed by centrifugation at 3000 rpm for 3 min. The remaining pellet containing the MPs were reconstituted in milli-Q water and washed three times by centrifugation at 21,000× *g* for 15 min. The final formulation was reconstituted in 1 mL of water. F1-F6 formulations were synthesized with minor alterations, including various polymers included in the inner aqueous phase (cationic polymer PEI and porogen NaHCO₃) and stabilizer PVA concentration. A summary of the formulation details is provided in Table 1.

### 4.4. Characterization of Cela MPs

#### 4.4.1. Physiochemical Characterization: Particle Size (PS), Polydispersity Index (PDI), and zeta Potential

A total of 20 µL of the formulation samples were separately diluted by 150× in HPLC water. Particle size (PS), polydispersity index (PDI), and zeta potential were measured using a Malvern Zetasizer (Malvern Panalytical Instruments Ltd., Malvern, UK).

#### 4.4.2. Drug Content

The percent encapsulation efficiency (%EE) and percent drug loading (%DL) of Cela in MPs were determined using a MP lysis method. Briefly, 20 µL of the formulation was lysed using a mixture of 10 µL DMSO, 20 µL DCM, and 1950 µL ACN, followed by centrifugation for 45 min at 21,191× *g* (4 °C). The supernatant containing free Cela was injected into the UPLC for analysis using the method defined in Section 2.2. The %EE and %DL were calculated based on Equations (1) and (2) respectively:(1)$$\%\mathrm{EE}=\left(\frac{\mathrm{Drug}\;\mathrm{entrapped}\;\mathrm{in}\;\mathrm{the}\;\mathrm{NPs}}{\mathrm{Initial}\;\mathrm{drug}\;\mathrm{added}}\right)\times 100\%\;\;\;\;\;\;\;\;\;\;\;\;\;\;\;\;\;\;\;\;\;\;\;\;\;\;\;\;\;\;\;$$
(2)$$\%\mathrm{DL}=\left(\frac{\mathrm{Drug}\;\mathrm{entrapped}\;\mathrm{in}\;\mathrm{the}\;\mathrm{NPs}}{\mathrm{Amount}\;\mathrm{of}\;\mathrm{PLGA}+\;\mathrm{Initial}\;\mathrm{Drug}\;\mathrm{added}}\right)\times 100\%\;\;\;\;\;\;\;\;\;\;\;\;\;\;\;\;\;\;\;$$

#### 4.4.3. Morphological Analysis

The morphology of Cela MPs was examined using Helios Nano Lab 660 (FEI, Hillsboro, Oregon, USA). The microparticle suspension was dried on the SEM pin stub (Ted Pella, Inc., Redding, CA, USA) and then sputter coated with gold for 90 s and imaged at 5 kV.

#### 4.4.4. In Vitro Release Study

Samples for an in vitro release study were prepared by diluting 50 µL Cela MPs with 950 µL of 1% sodium lauryl sulfate (SLS) in Simulated Lung Fluid (SLF), which was prepared according to the components in Gamble’s Solution [82]. The samples were shaken and incubated at 200 rpm and 37 °C with an incubating orbital shaker. At each time point between 0.5–72 h, one sample was removed and centrifuged at 17,500× *g* for 15 min. The supernatant was then analyzed by UPLC, as described in Section 2.2.

#### 4.4.5. Differential Scanning Calorimetry (DSC) Studies

Various samples were first lyophilized into dry powders using a Labconco FreeZone^®^ freeze dryer system. DSC analyses were performed on a DSC 6000 (PerkinElmer, Inc; Waltham, MA, USA), using samples of 2–5 mg inside sealed pans. The samples were heated in sealed aluminum pans at a rate of 10 °C/min from 25–250 °C under a nitrogen flow rate of 50 mL/min. A blank sealed aluminum pan was used as the reference.

#### 4.4.6. X-ray Diffraction Studies (XRD)

XRD analysis was performed using XRD-6000 (Shimadzu, Kyoto, Japan). The diffraction patterns were measured with a voltage of 40 kV and a current of 30 mA. The freeze-dried samples were uniformly spread on a glass sample holder and analyzed over a range of 2θ or 10–60° at a scan speed of 2 °/min.

### 4.5. In Vitro Stability Study

Triplicate samples of Cela MP suspensions were stored at 4 °C for a 4-week period. A total of 20 µL aliquots of each sample were withdrawn each week to determine the PS, PDI, and zeta potential by a Malvern Zetasizer as previously described. The %EE was analyzed by UPLC.

### 4.6. Aerosolization, Aerodynamic Properties, and Inhalability of Cela MPs

The in vitro aerodynamic assessment of Cela MPs was performed using the Copley^®^ 170 Next Generation Impactor^TM^ (NGI, MSP Corporation, Shoreview, MN, USA), following a previous report [58]. Briefly, the NGI plates were refrigerated for 2 h at 4 °C to prevent thermal transfer within the cascade impactor [52]. A total of 2 mL of Cela MP suspension was loaded into the nebulizer cup, which was attached to a PARI^®^ LC plus nebulizer (PARI Respiratory Equipment, Midlothian, VA, USA). The vacuum pump (MSP Corp, Shoreview, MN, USA) was set at a flow rate of 15 L/min for 4 min as the sample travelled into the NGI. After the run, ACN was used to wash and collect the samples from each stage plate, followed by centrifugation at 21,191× *g* for 45 min to lyse the MPs. The resulting supernatant was analyzed using the UPLC method as mentioned in Section 2.2. PS, PDI, and zeta potential of Cela MPs were evaluated after nebulization to test whether the MPs maintained stability.

The Fine Particle Fraction (FPF, %) was calculated as the ratio of fine particle dose (d_ae_ < 5.39 μm or amount of drug deposited from stage 3–8) to the total emitted dose (ED) (amount of drug emitted from mouthpiece to stage 8) deposited in the NGI. The mass median aerodynamic diameter (MMAD, D_50%_) was determined by acquiring the diameter corresponding to 50% of the cumulative deposition. The geometric standard deviation (GSD) was calculated using Equation (3) as shown below:(3)$$\mathrm{GSD}=\left(\frac{D^{84.1\%}}{D^{15.9\%}}\right)\;\;\;\;$$

### 4.7. In Vitro Cell Culture Studies for the Determination of Anti-Cancer Efficacy

The present work was carried out using four malignant pleural mesothelioma cell lines, MSTO-211H, H28, H2452, and primary ROB cells [4,83]. These cells were chosen as ideal models for mesothelioma, as they were derived directly from patient tumors, and therefore accurately represented in vivo tumor cells.

#### Cytotoxicity Studies

Cell viability studies for Cela and Cela MP were evaluated against MSTO-211H, H28, and H2452 cells, in addition to using a primary patient-derived cell line ROB using an MTT (3-(4,5-dimethylthiazol-2-yl)-2,5-diphenyl tetrazolium bromide) assay, following the methods published in our earlier studies [84,85]. Detailed methods are provided in Appendix A. IC_50_ values were calculated using the non-linear fitting module available in GraphPad Prism software (Version 6.0 for Windows, GraphPad Software, CA, USA). To further evaluate the toxicity of blank MPs, toxicity studies were performed in MSTO-211H and NHLF cell lines at corresponding celastrol concentrations (0.10–6.25 µM) after a 48-h incubation period.

### 4.8. Mechanistic Studies

#### 4.8.1. DPPH Antioxidant Assay

The potent antioxidant and anti-inflammatory activity of Cela has been shown to be useful to treat a multitude of diseases such as cancers (e.g., glioma, ovarian cancer, and colorectal cancer) [79,86,87,88,89]. 2,2-Diphenyl-1-picrylhydrazyl (DPPH) is a free radical that produces a violet solution once dissolved in methanol, and in the presence of an antioxidant, the free radical reduces to produce a colorless solution. Therefore, the DPPH assay was selected to evaluate the antioxidant capacity, or free radical scavenging ability, of Cela and Cela MP based on a previously described method [71]. Detailed methods are provided in Appendix A.

#### 4.8.2. Caspase-3 Assay

To evaluate the apoptotic properties of Cela and Cela MPs, the apoptosis mediator (caspase-3) activity was determined using an EnzChek Caspase-3 assay kit (Thermo Fisher Scientific, Waltham, MA, USA). MSTO-211H cells were seeded at a density of 5 × 10^5^ cells/TC dish (100 mm diameter) (Thermo Scientific, Rochester, NY, USA). Treatments of Cela and Cela MP (0.5 and 1.0 μM) were incubated for 6 h, after which cells were harvested and washed with sterile PBS, followed by resuspending in 1X cell-lysis buffer and centrifugation. A total of 50 μL of the supernatant and 50 μL of a 2X substrate working solution (10 mM Z-DEVD-AMC substrate + 2X reaction buffer) were added to a 96-well plate while using 50 μL of lysis buffer, and a substrate working solution was added as a no enzyme control. The fluorescence was measured after incubating for 20 min at excitation/emission of 360/460 nm.

#### 4.8.3. Effect of Cela and Cela MP on Cellular Autophagy

The CYTO-ID^®^ Autophagy Detection Kit (Enzo Life Sciences, Farmingdale, NY, USA) was used to determine the effect of Cela and Cela MP on autophagy in MPM cells by following a previously published protocol [4]. MSTO-211H cells were used for this study. Detailed methods are provided in Appendix A.

### 4.9. Determination of Cela MP Efficacy in 3D Tumor Spheroid Model

The anti-cancer efficacy of Cela MPs was further evaluated by determining the penetrability and efficacy of nanoparticles into solid tumors. As previous reports have indicated, 3D spheroid cell culture studies have been used in an attempt to bridge the gap between in vitro and in vivo systems by partially representing the tumor structure and microenvironment [90,91,92]. Our earlier study has suggested the feasibility of MSTO-211H 3D spheroids in testing efficacy of various therapeutics [93]. Briefly, MSTO-211H cells were seeded in a Corning^®^ ultra-low attachment spheroid 96-well plate (Corning, NY, USA) at a density of 2.0 × 10^3^ cells/well and incubated at 37 °C/5% CO_2_ for 3 days to allow spheroids to grow into a solid tumor mass, following which treatments were started (Day 0 for treatment). Totals of 0.5 and 1.0 µM celastrol concentrations were selected for treatment. For a single dose study, half the media (100 µL) was replaced with fresh media on each imaging day (except for Day 0). For the multiple dose study, half of the media was replaced with respective concentrations of Cela or Cela MP or fresh media (control). Images of the spheroids were taken using an inverted microscope (10× magnification, Laxco LMI-6000, Laxco Inc., Mill Creek, WA, USA) on days 0, 3, 6, 9, 12, and 15. NIH ImageJ software (Version 1.44) was used to measure the diameter of the spheroids, which was then used to calculate the spheroid volumes.

#### 4.9.1. Cell Titer-Glo Cell Viability Study

At the end of Day 15, cell viability within the spheroid core was assessed using Cell Titer-Glo^®^ kit (Promega, Madison, WI, USA). Briefly, 100 µL of treatment was removed and replaced with 100 µL of CellTiter-Glo^®^ reagent in each well. The contents were mixed for a few minutes, followed by incubation at room temperature for 30 min. Luminescence was measured using a Spark 10M plate reader (Tecan, Männedorf, Switzerland).

#### 4.9.2. Live/Dead Cell Assay

A Live/Dead cell assay was performed using a fluorescent Viability/Cytotoxicity assay kit (Biotium, Fremont, CA, USA) to visualize the live and dead cells on the surface of the spheroids. At day 15, the treatments were completely removed from each well and replaced with 100 µL of 2 µM calcein AM/4 µM EthD-III staining solution. The plate was then incubated in the dark at room temperature for 30 min and subsequently imaged on an EVOS-FL Cell Imaging fluorescence microscopy at 4X magnification (Thermo Fisher Scientific, Waltham, MA, USA). Green fluorescent protein (GFP, imaging calcein AM) is representative of viable cells, whereas red fluorescent protein (RFP, imaging EthD-III) indicates necrotic/dead cells. Fluorescent intensity was analyzed by NIH ImageJ software (Version 1.53c).

### 4.10. Statistical Analyses

All data were addressed as mean ± SD or SEM, with n = 3 unless otherwise mentioned. At least three trials of cytotoxicity studies were performed for each control or treatment with n = 6 for each trial. All data were evaluated by an unpaired student’s *t*-test or one-way ANOVA followed by Tukey’s multiple comparisons test using GraphPad Prism software (Version 6.0 for Windows, GraphPad Inc., San Diego, CA, USA). A *p* value of <0.05 was considered statistically significant and was presented in data figures as a single asterisk (*). However, some studies demonstrated a smaller *p*-value of 0.01 or less, which is indicated at respective places.

## 5. Conclusions

In this study, surface-wrinkled Cela-loaded microparticles were developed with a suitable aerodynamic diameter for efficient deep lung delivery. Cela MP was able to retain the antioxidant activity of the drug, trigger autophagy, and induce apoptosis. These mechanisms, in combination with an efficient delivery system, resulted in significantly enhanced efficacy against MPM in both 2D cell culture and a 3D tumor spheroid model. This study lays the groundwork to pursue the development of Celastrol and Celastrol-encapsulated delivery systems in preclinical and potentially clinical settings. These findings also suggest that the successful encapsulation of Cela in polymeric microparticles addresses certain limitations of Cela, such as low aqueous solubility and potential toxicity at higher doses. Going forward, we plan to study the benefits of Cela MPs in vivo and evaluate their long-term safety and stability profiles. In conclusion, inhaled Cela MPs are a promising anti-cancer standalone therapy for MPM treatment.

## Figures and Tables

**Figure 1 ijms-24-05204-f001:**
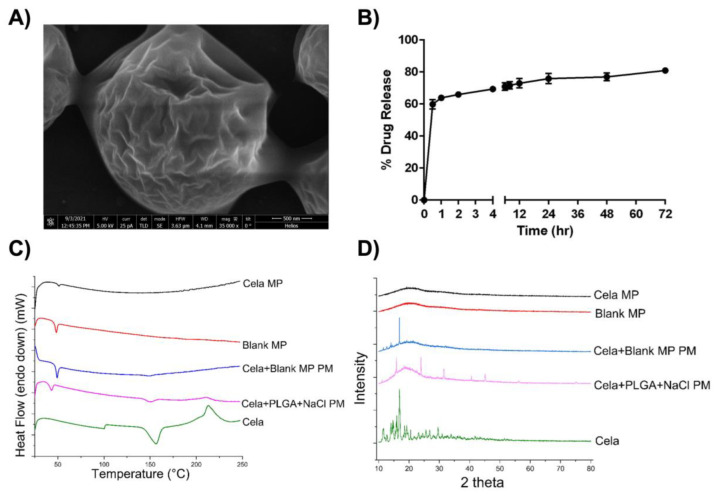
Imaging and physical characterizations of Cela MP. (**A**). SEM image showing a wrinkled surface with a ~2 µm diameter of Cela MP. (**B**). In vitro drug release of Cela MP in SLF with 1% SLS. Data represent mean ± SD (n = 3). (**C**). Overlay of DSC thermograms. (**D**). Overlay of XRD patterns of Cela, Cela + PLGA + NaCl physical mixture, Cela + Blank MP physical mixture, Blank MP, and Cela MP.

**Figure 2 ijms-24-05204-f002:**
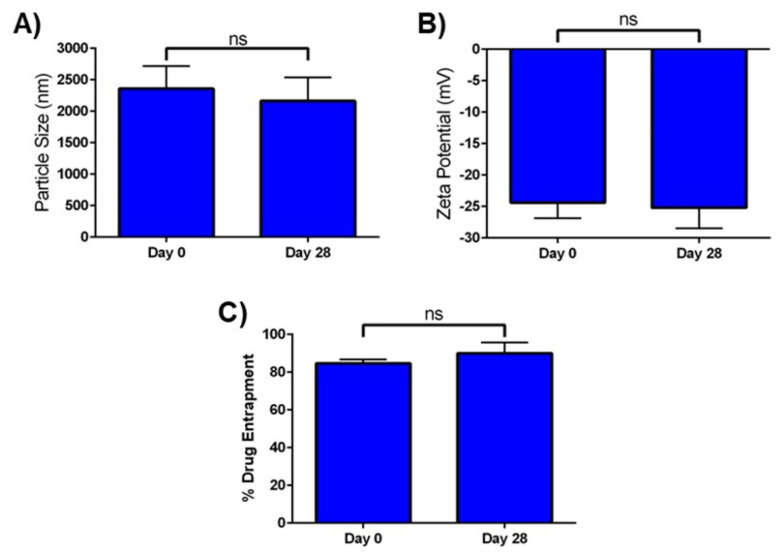
Stability studies of Cela MPs over a span of 28 days at 4 °C measuring (**A**). Particle size (nm), (**B**). Zeta potential (mV), and (**C**). percent drug entrapment. Data represent mean ± SD (n *=* 3). ns = not significant.

**Figure 3 ijms-24-05204-f003:**
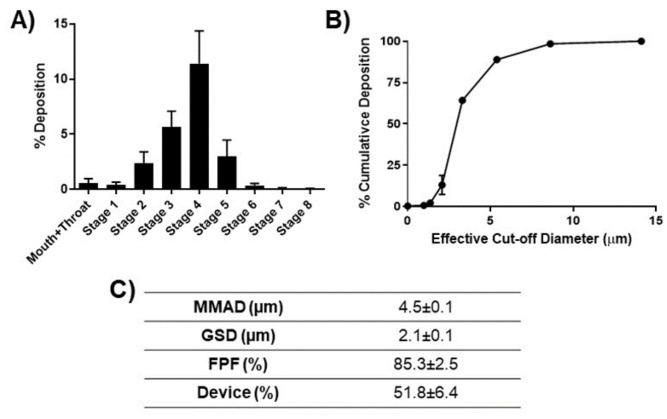
In vitro aerosol performance of Cela MP. (**A**). Aerodynamic deposition of Cela MPs in various stages of a Next-Gen Impactor^TM^ (NGI). (**B**). Percent cumulative deposition as a function of effective cut-off diameter of Cela MPs. (**C**). Summary of aerosolization parameters calculated from using NGI. Data represent mean ± SEM (n = 3).

**Figure 4 ijms-24-05204-f004:**
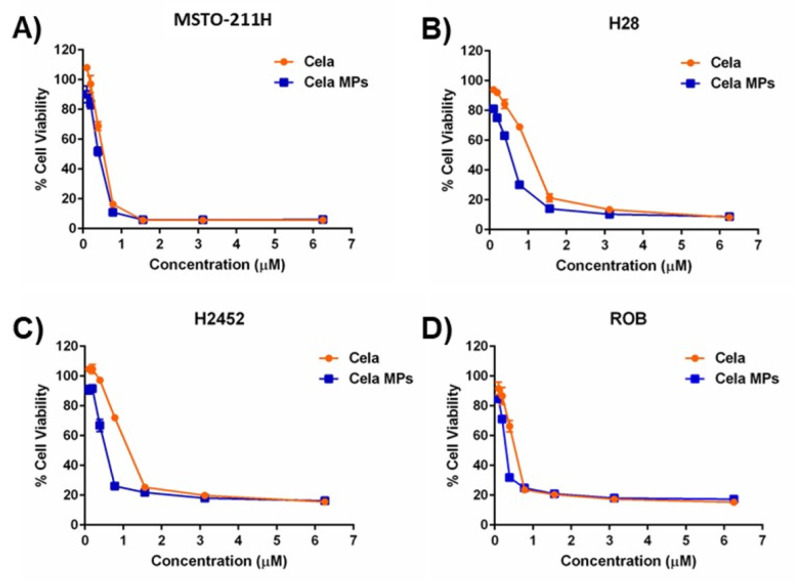
Cytotoxicity of Cela and Cela MPs against (**A**). MSTO-211H, (**B**). H28, (**C**). H2452, and (**D**). ROB cell lines. Cells were treated with different concentrations of each treatment for 48 h and cell viability was measured using MTT assay. Cells grown in media were considered as control (100%). Data represent mean ± SD of three separate trials, each with n = 6.

**Figure 5 ijms-24-05204-f005:**
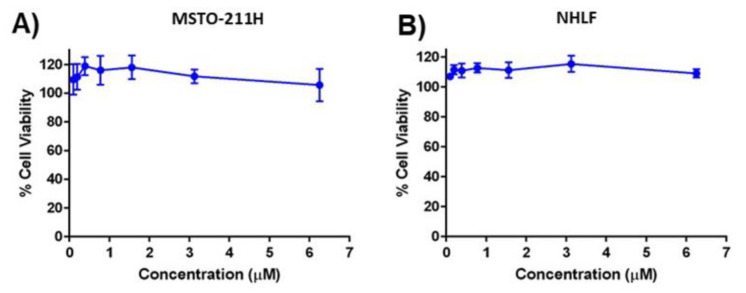
Toxicity study for (**A**). MSTO-211H and (**B**). Normal human lung fibroblast (NHLF) cells. Cells were treated with different concentrations of each treatment for 48 h and the cell viability was measured using an MTT assay. Cells grown in media were considered as the control (100%). Data represent mean ± SD (n = 6).

**Figure 6 ijms-24-05204-f006:**
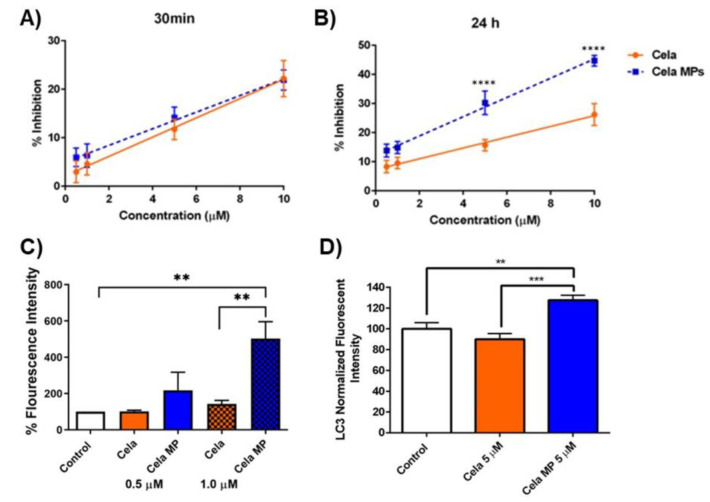
Maintenance of antioxidant properties of Cela MPs and activation of apoptosis and autophagy pathways. (**A**). & (**B**). DPPH radical scavenging activity of different concentrations of Cela MPs (0.5–10 µM) when compared with Cela measured and calculated at 30 min (**A**) and 24 h (**B**). Data represent mean ± SD (n = 3). Significance between the groups was analyzed by multiple *t*-test analyses. Data represent mean ± SD (n = 3). (**C**). Quantification of caspase-3 as percent fluorescence intensity of Cela and Cela MP treatments as compared to control on MSTO-211H cells after 6 h treatment. Data represent mean ± SEM (n = 3). (**D**). Representation of autophagy induction on MSTO-211H cells by measuring LC3B protein fluorescence intensity (normalized). Data represent mean ± SD (n = 3). *** p* < 0.01, **** p* < 0.001, and ***** p* < 0.0001.

**Figure 7 ijms-24-05204-f007:**
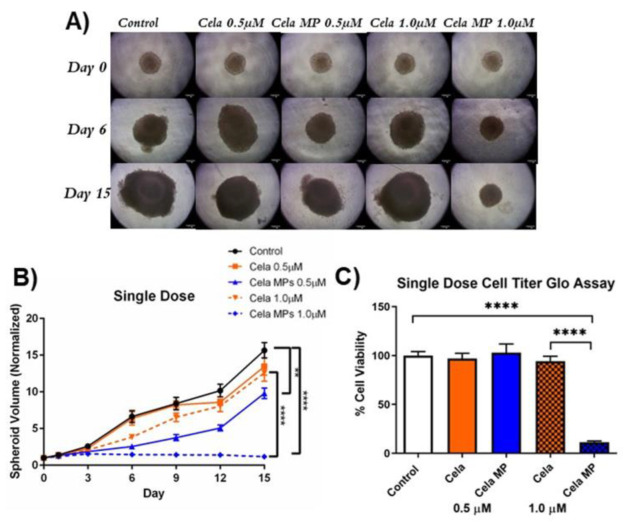
Single-dose 3D spheroid assay to determine the effects of treatment on MSTO-211H tumor spheroid growth. (**A**). Representative images of single-dose study taken at 10X magnification of spheroids on days 0, 3, 9, and 15. Scale bar represent 500 µm. (**B**). Quantification of spheroid volume (normalized to day 0 spheroid volume) over 15 days for single-dose study. Data represent mean ± SEM (n = 8). (**C**). Single-dose 3D cell viability study was performed using CellTiter^®^ Assay. Data represent mean ± SD (n = 4). Significance between groups was analyzed by one-way ANOVA and *Tukey’s* multiple comparisons test. ** *p* < 0.01 and ***** p* < 0.0001.

**Figure 8 ijms-24-05204-f008:**
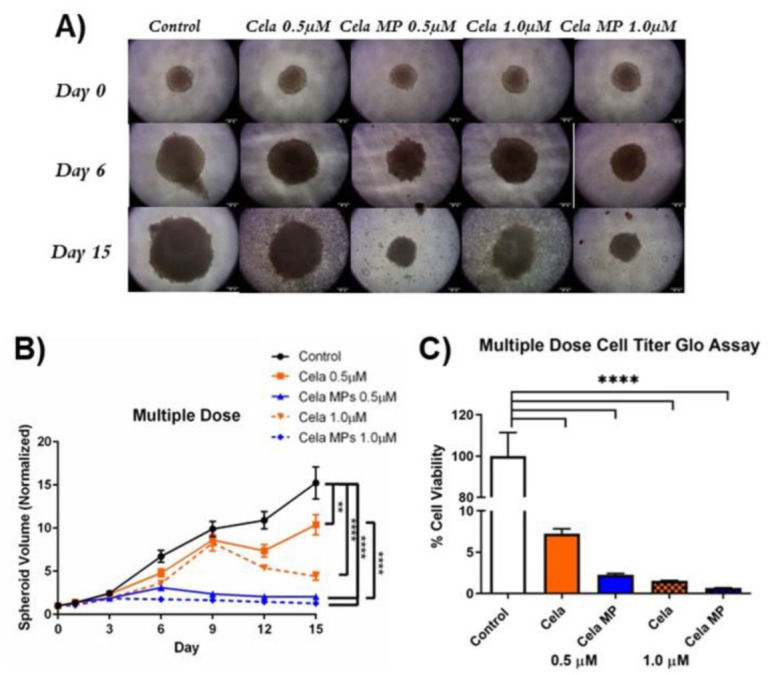
Multiple-dose 3D spheroid assay to determine the effects of treatment on MSTO-211H tumor spheroid growth. (**A**). Representative images of multiple-dose study taken at 10X magnification of spheroids on days 0, 3, 9, and 15. Scale bar represent 500 µm. (**B**). Quantification of spheroid volume (normalized to day 0 spheroid volume) over 15 days for multiple-dose study. Data represent mean ± SEM (n = 8). (**C**). Multiple-dose 3D cell viability study was performed using CellTiter^®^ Assay. Data represent mean ± SD (n = 4). Significance between groups was analyzed by one-way ANOVA and *Tukey’s* multiple comparisons test. ** *p* < 0.01 and ***** p* < 0.0001.

**Figure 9 ijms-24-05204-f009:**
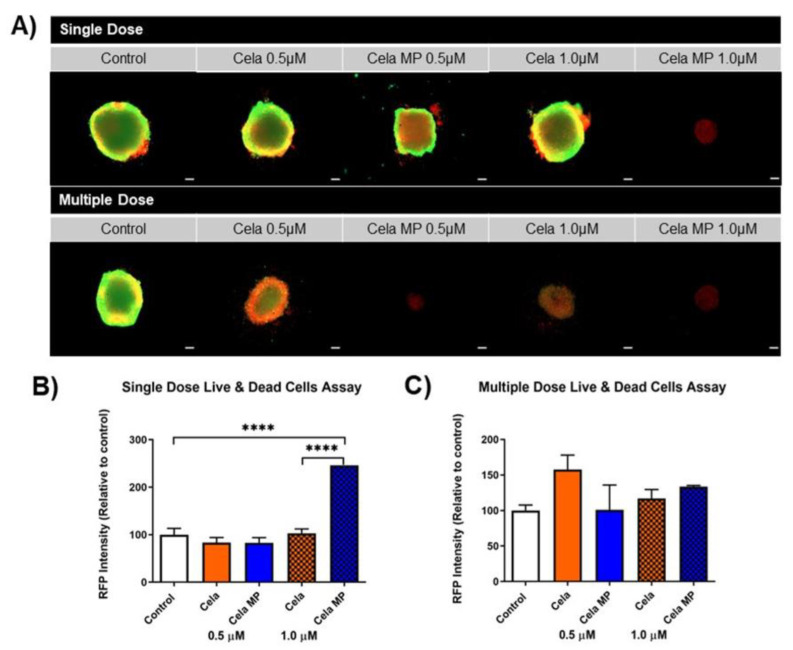
Live/Dead cell assay performed using Viability/Cytotoxicity Assay Kit for Animal Live & Dead Cells on day 15 MSTO-211H 3D spheroids. (**A**). Representative images (4× magnification) of stained spheroids for both single and multiple-dose 3D spheroid studies. GFP depicts live cells while RFP depicts compromised/dead cells. Scale bar for the images represents 500 μm. Quantitative representations of percent RFP fluorescence intensity of each treatment for (**B**). single-dose and (**C**). multiple-dose 3D spheroid study. Control was considered 100%. Data represent mean ± SEM (n = 3). Significance between groups was analyzed by one-way ANOVA and *Tukey’s* multiple comparisons test. **** *p* < 0.0001.

**Table 1 ijms-24-05204-t001:** Methods and physical characterizations of optimized formulation of Cela MPs (F6). Data represent mean ± SD (n = 3).

No.	A1	O1	A2	% Entrapment Efficiency	Particle Size (nm)	PDI	Zeta Potential (mV)
F6	Water + 1% NaCl	2 mg drug in DCM and DMSO	1% PVA	72.8 ± 6.1	2076 ± 390	0.3 ± 0.2	−36.6 ± 3.8

A1—Inner aqueous phase. O1—Organic phase. A2—Outer aqueous phase.

**Table 2 ijms-24-05204-t002:** Quantification and comparison of IC_50_ values derived from cytotoxicity studies. Data represent mean ± SD of three independent trials of n = 6 for each trial. Significance between groups was analyzed by an unpaired *t*-test. * *p* < 0.05, ** *p* < 0.01, **** *p* < 0.0001.

Cell Line	IC_50_ (µM)	
	Cela	Cela MP
MSTO-211H	3.6 ± 0.4	2.7 ± 0.3 *
H28	12.9 ± 3.3	3.8 ± 0.6 **
H2452	18.7 ± 0.1	3.8 ± 0.1 ****
ROB	3.5 ± 0.2	2.3 ± 0.3 **

## Data Availability

The data shown in this study are presented in the article. Also, the data presented in this study are available on request from the corresponding authors.

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
