# Peer review of "Surface-Modified Inhaled Microparticle-Encapsulated Celastrol for Enhanced Efficacy in Malignant Pleural Mesothelioma"

_ijms, 2023, doi:10.3390/ijms24065204_

Round 1
Reviewer 1 Report
1) In section 3.1, what's the criteria of determine a high %EE? 63.7% may not be claimed as a high %EE. Instead, the author can claim that this is relatively higher than other particles prepared. I think the author increased zeta potential by adding PEI, instead of "to reduce the surface charge". Please clarify. What's the rationale behind inclusion of PEI and what's the reason that eventually PEI was not included in F6 formulation?
2) In Table 1, results for F6 were reported with standard deviation (+/-), but results for F1-F5 were not. Why is that? Please keep consistent.
3) The authors has performed in vitro assays including spheroids assay, but not in vivo study yet. Therefore, I would not suggest using "tumor growth inhibition" in the abstract and discussion, because they are in vitro spheroids, not tumors. To be precise, the author can rephrase as "growth inhibition in spheroids".
4) How will the final formulation be administered by inhalation, by nasal dripping or aerosol? Can the MP maintain its stability after being nebulized? I'm looking forward to hearing authors' thoughts/plan on the approach of administration.
5) Duplicated 2.4.3, please correct.
Author Response
Reviewer 1
- In section 3.1, what's the criteria of determining a high %EE? 63.7% may not be claimed as a high %EE. Instead, the author can claim that this is relatively higher than other particles prepared. I think the author increased zeta potential by adding PEI, instead of "to reduce the surface charge". Please clarify. What's the rationale behind inclusion of PEI and what's the reason that eventually PEI was not included in F6 formulation?
Author Response: Thank you for your comment. We have reworded the description for 63.7% as “relatively higher” instead of “high” %EE for clarification (Page 16). Edits were made to clarify PEI addition as a method to increase zeta potential instead of “to reduce the surface charge” (Page 15). PEI was originally included in the formulation to increase zeta potential as that may lead to improved cellular uptake. However, PEI inclusion had a negative impact on %EE (F2 and F4 formulations). Additionally, F2 formulation resulted in high PDI, possibly due to lack of uniformity in particle sizes and aggregation. Therefore, we did not include PEI for further formulations. A statement is now added to the manuscript as an explanation (Page 16).
- In Table 1, results for F6 were reported with standard deviation (+/-), but results for F1-F5 were not. Why is that? Please keep consistent.
Author Response: Thank you for your suggestion. During formulation optimization process, only one trial was performed for F1-F5 formulations, therefore no standard deviations are reported. Once F6 was optimized with the most ideal characteristics, we then repeated the formulations (n=3). This is consistent with some of our previous published works (1,2). To remove confusion, we have moved the characterization table for F1-F5 to the supplementary (Table S1) and have updated Table 1 with information on only the optimized F6 formulation.
- The authors have performed in vitro assays including spheroids assay, but not in vivo study yet. Therefore, I would not suggest using "tumor growth inhibition" in the abstract and discussion, because they are in vitro spheroids, not tumors. To be precise, the author can rephrase as "growth inhibition in spheroids".
Author Response: Thank you for your correction. We have made appropriate changes to replace “tumor” with “spheroid” in the abstract (Page 3), results (Pages 21 and 22), and discussion (Page 28) sections.
- How will the final formulation be administered by inhalation, by nasal dripping or aerosol? Can the MP maintain its stability after being nebulized? I'm looking forward to hearing authors' thoughts/plan on the approach of administration.
Author Response: The final formulation will be administered by mouth inhalation via nebulization of the microparticle suspension. In this project, we tested a marketed jet nebulizer, PARI® LC plus nebulizer, for in-vitro lung deposition study. From this study, we can conclude that Cela MPs possessed good aerodynamic behaviors and can be readily delivered to the deep lung region to treat the diseased site. As for the concern over the MP stability after nebulization, we have added an additional study to evaluate %EE, PS, PDI, and zeta potential of Cela MPs after nebulization (Page 11). From the results of this study, Cela MPs displayed insignificant changes in all the four parameters tested, therefore confirming their stability after nebulization. Corresponding information has been added to the manuscript on Pages 11, 18, and 26; and also, as supplementary data, Table S2.
- Duplicated 2.4.3, please correct.
Author Response: Thank you for your comment. We have checked the manuscript but did not find a duplicated 2.4.3. All other sections and subsections were double-checked as well. Please let us know if you meant something else.
Reviewer 2 Report
The authors of the article “Surface-Modified Inhaled Microparticle-encapsulated Celastrol for Enhanced Efficacy in Malignant Pleural Mesothelioma” developed a methodology for obtaining surface-modified Cela-loaded microparticles for the treatment of Malignant pleural mesothelioma using a double emulsion solvent evaporation method. The reported issues are relevant in the field of nanotechnology and for exploring new approaches to drug delivery systems.
This manuscript can be considered for publication in this Journal after minor revision. In general, many typical mistakes (grammar, unclear sentences)
Author Response
Reviewer 2
- The authors of the article “Surface-Modified Inhaled Microparticle-encapsulated Celastrol for Enhanced Efficacy in Malignant Pleural Mesothelioma” developed a methodology for obtaining surface-modified Cela-loaded microparticles for the treatment of Malignant pleural mesothelioma using a double emulsion solvent evaporation method. The reported issues are relevant in the field of nanotechnology and for exploring new approaches to drug delivery systems. This manuscript can be considered for publication in this Journal after minor revision. In general, many typical mistakes (grammar, unclear sentences)
Author Response: Thank you for your valuable suggestion. We have thoroughly proofread the manuscript and have made corrections for any grammatical and other errors.
Reviewer 3 Report
-It must be well-discussed why Cela MP are significantly more effective than free Cela. Why NP form has better cell attachment?
- For safey studies, it is also necessary to evaluate the cytotoxic effect of Cela MP.
-This work does not have novelty as particles are previously studied for pulmonary drug delivery.
- The effectiveness of the pulmonary delivery of this system compared to other common routes must be proved.
-Mucoadhesiveness of the particles must be demonstrated.
-Since clearance by macrophage is an important barrier, overcoming this barrier must be shown.
-Overall, the current spheroid Model cannot represent in vivo systems in many ways and it cannot warrant same outcome in animal model.
Author Response
Reviewer 3
- It must be well-discussed why Cela MP are significantly more effective than free Cela. Why NP form has better cell attachment?
Author Response: Thank you for your comment. We have added more discussion on why Cela MP performed significantly better than free Cela, as well as the explanation on better cell attachment (Page 26). In the discussion section, we mentioned the superior efficacy of Cela MP was due to better cell attachment, that may cause enhanced cellular uptake and subsequent cytotoxicity. In nature, particles with wrinkled surfaces are widely found such as plant pollens, plant sees, and microorganisms (i.e., neutrophils). These wrinkles with their significantly enlarged surface areas provide enhanced survival tools including pollen adhesion and hydration, and cell signaling (3,4). Inspired by these irregular morphologies, a study by Li et al. prepared wrinkled particles that cells readily attached, climbed and conformed onto the particles without any chemical modifications, which was not observed on the smooth particles (5). Cell attachment to the surface of the particles was observed with actin networks appearing at the particle edges (5). These findings provide an explanation for our observations of Cela MPs performing significantly better than free Cela. In addition, in this project, we proposed the route of administration was via inhalation for local delivery of therapeutics. However, simple pulmonary delivery of free drug suspension is insufficient and unpredictable for deep lung deposition due to the large crystalline drug particles and variability in particle sizes (Page 25). On the other hand, Cela MPs were characterized with uniform particle size and ideal aerodynamic properties and can provide additional protection of the free drug during nebulization. Multiple studies have demonstrated the wrinkle morphology enhances the aerodynamics of aerosol in inhalation purposes (6–8). Thus, an improved deposition in the target lung tissues would also contribute to the enhanced therapeutic efficacy of Cela MP.
- For safety studies, it is also necessary to evaluate the cytotoxic effect of Cela MP.
Author Response: Thank you for your correction. We have actually performed these studies to test the toxicity of blank MPs to ensure the cytotoxic effects are due to Cela encapsulation into the MPs and not from other formulation components. Results from our study demonstrated that the blank MPs displayed negligible toxicity on the cell lines tested, thus confirming cytotoxic effects are indeed due to MP encapsulation. These corrections have been made on Pages 13, 20, and 27.
- This work does not have novelty as particles are previously studied for pulmonary drug delivery.
Author Response: Thank you for your comment. However, we disagree with the reviewer regarding the lack of novelty for these particles for pulmonary drug delivery. We did a quick search on PubMed with keywords, “wrinkled particle” and “inhalation” and found 27 results. All the published wrinkled particles were developed into dry powder inhalers and the majority of them were prepared using dry-spraying technique. Whereas, in this project, we prepared wrinkled microparticles by adding a pore-forming agent during the double emulsion solvent evaporation technique, therefore the preparation method is novel. In addition, we proposed pulmonary delivery via nebulization of microparticles in suspension, which is also very different from dry powder inhalers. In addition, none of the above-mentioned studies has tested efficacy of inhaled delivery systems for treatment of malignant pleural mesothelioma (MPM), a rare cancer of the lungs’ pleural cavity. Taken together, a novel approach was used to develop wrinkled particles for nebulized pulmonary drug delivery to treat a rare cancer of the respiratory system.
- The effectiveness of the pulmonary delivery of this system compared to other common routes must be proved.
Author Response: Thank you for your suggestion. We have added additional discussion to compare conventional routes of administrations such as oral and intravenous to inhalation (Page 25-16). In the manuscript, we have mentioned that conventional routes of administration such as oral and intravenous are not ideal for Cela delivery. For instance, pharmacokinetic study done on rats demonstrated the mean oral absolute bioavailability after oral administration (3mg/kg) was very low at 3.14% (9). Even though bioavailability is not a concern in intravenous administration, Cela is associated with many side effects including infertility, cardiotoxicity, hepatotoxicity, hematopoietic system toxicity, and nephrotoxicity (10). Thus, pulmonary route of administration is inarguably the best choice to efficiently deliver therapeutics locally to the lungs to improve local bioavailability and avoid adverse effects. In a recent report, inhaled nintedanib (given at 1:120 of the oral dose) was found to deliver an oral-equivalent lung Cmax with lower systemic AUC, was well-tolerated and effective at reducing bleomycin-induced pulmonary fibrosis (11). Therefore, we believe pulmonary delivery is the most ideal route of administration for Cela MP for the treatment of a rare cancer of the respiratory system.
- Mucoadhesiveness of the particles must be demonstrated.
Author Response: Thank you for your comment. However, we believe mucoadhesiveness of the particles would not be ideal for the purpose of this project. Mucus is generally located at the upper airways of the lungs. Thus, mucociliary clearance acts as the primary defense mechanism in this region (12). If the aim was to deliver particles in the upper airways, then mucoadhesive properties would be necessary to be investigated. However, the desired deposition of Cela MPs are within the peripheral lungs, in alveolar region. In the deep lung region, there are almost no mucus present to ensure proper air exchange to occur. Therefore, mucoadhesiveness is not a desired characteristic for Cela MPs in this project.
- Since clearance by macrophage is an important barrier, overcoming this barrier must be shown.
Author Response: Thank you for your suggestion. We understood the importance of macrophage clearance would influence the fate of Cela MPs, so cellular uptake studies in mesothelioma cells and macrophages were attempted in this project. Briefly, MPs were prepared by replacing Cela with coumarin-6, a fluorescent dye that has been well established in uptake studies (13,14). MSTO-211H and macrophage cells were incubated with plain coumarin-6 or coumarin-6 MPs (both at 1 µg/mL concentration) for 3 h, and then the cells were fixed and DAPI nuclear stain was added. A similar quantification study was performed by obtaining the cell pellets after treatments, and the green fluorescence (GFP) was detected using Cellometer® Vision Cell Profiler (Nexcelom Bioscience). In both uptake studies, there were significant signal interference coming from the MPs that made evaluation impossible for the Cellometer. This is because the micron size of the MPs overwhelms the proper detection of individual cells that were only a few microns bigger. As a result, the data from the cellular uptake studies were inconclusive and thus were not included in the manuscript. In the manuscript, we have mentioned a few studies describing changes in surface morphology can circumvent uptake by macrophages (Pages 7 and 24). For future studies, we would like to explore alternative methods to evaluate whether Cela MPs can avoid macrophage uptake.
- Overall, the current spheroid Model cannot represent in vivo systems in many ways and it cannot warrant same outcome in animal model.
Author Response: Thank you for your comment. We understand that the 3D spheroid study cannot represent in-vivo systems. We specifically mentioned in the manuscript that “the efficacy of Cela MP was evaluated using a 3D spheroid model to mimic in-vivo conditions” (Page 29). We have also added an additional statement mentioning this study acts to bridge the gap between in-vitro and in-vivo studies, to allow for a more accurate representation. The main aim of the present study was to prepare a wrinkled surface microparticle for the encapsulation of Cela and investigate its inhalation feasibility and anti-cancer efficacy against malignant mesothelioma, which was successfully proven. The preliminary 3D spheroid study gives us an idea of what would happen in-vivo. A whole separate study exploring the efficacy of inhaled Cela MP in mesothelioma animal model would be appropriate in this case.
Round 2
Reviewer 3 Report
This work does not have enough novelty and the modifications do not make it acceptable.